# SOSELETO: A Unified Approach to Transfer Learning and Training with Noisy Labels

## Abstract

We present SOSELETO (SOurce SELEction for Target Optimization), a new method for exploiting a source dataset to solve a classification problem on a target dataset. SOSELETO is based on the following simple intuition: some source examples are more informative than others for the target problem. To capture this intuition, source samples are each given weights; these weights are solved for jointly with the source and target classification problems via a bilevel optimization scheme. The target therefore gets to choose the source samples which are most informative for its own classification task. Furthermore, the bilevel nature of the optimization acts as a kind of regularization on the target, mitigating overfitting. SOSELETO may be applied to both classic transfer learning, as well as the problem of training on datasets with noisy labels; we show state of the art results on both of these problems.

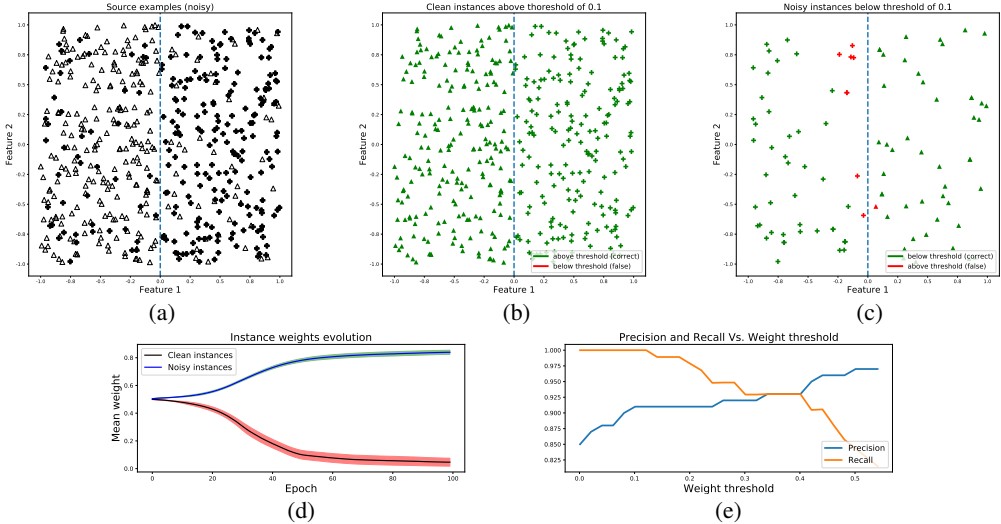

Figure 1: SOSELETO applied to a synthetic noisy labels problem. (a) A binary classification problem with points split by the y-axis. Input labels are marked as diamonds and triangles. 20% of the input labels are noisy (have the wrong label). SOSELETO assigns a weight per instance. (b) All correctly labeled input points are assigned high weights. (c) Most noisy points are assigned a low weight. (d) Mean weight of clean and noisy instances throughout training. (e) High accuracy and high recall are achieved for a broad range of weight thresholds.

## 1 Introduction

Deep learning has made possible many remarkable successes, leading to state of the art algorithms in computer vision, speech and audio, and natural language processing. A key ingredient in this success has been the availability of large datasets. While such datasets are common in certain settings, in other scenarios this is not true. Examples of the latter include "specialist" scenarios, for instance

a dataset which is entirely composed of different species of tree; and medical imaging, in which datasets on the order of hundreds to a thousand are common.

A natural question is then how one may apply the techniques of deep learning within these relatively data-poor regimes. A standard approach involves the concept of *transfer learning*: one uses knowledge gleaned from the *source* (data-rich regime), and transfers it over to the *target* (data-poor regime). One of the most common versions of this approach involves a two-stage technique. In the first stage, a network is trained on the source classification task; in the second stage, this network is adapted to the target classification task. There are two variants for this second stage. In feature extraction (e.g. Donahue et al. (2014)), only the parameters of the last layer (i.e. the classifier) are allowed to adapt to the target classification task; whereas in fine-tuning (e.g. Girshick et al. (2014)), the parameters of all of the network layers (i.e. both the features/representation and the classifier) are allowed to adapt. The idea is that by pre-training the network on the source data, a useful feature representation may be learned, which may then be recycled – either partially or completely – for the target regime. This two-stage approach has been quite popular, and works reasonably well on a variety of applications.

Despite this success, we claim that the two-stage approach misses an essential insight: *some source examples are more informative than others for the target classification problem.* For example, if the source is a large set of natural images and the target consists exclusively of cars, then we might expect that source images of cars, trucks, and motorcycles might be more relevant for the target task than, say, spoons. However, this example is merely illustrative; in practice, the source and target datasets may have no overlapping classes at all. As a result, we don't know *a priori* which source examples will be important. Thus, we propose to learn this source filtering as part of an end-to-end training process.

The resulting algorithm is SOSELETO: SOurce SELEction for Target Optimization. Each training sample in the source dataset is given a weight, corresponding to how important it is. The shared source/target representation is then optimized by means of a bilevel optimization. In the interior level, the source minimizes its classification loss with respect to the representation parameters, for fixed values of the sample weights. In the exterior level, the target minimizes its classification loss with respect to both the source sample weights and its own classification layer. The sample weights implicitly control the representation through the interior level. The target therefore gets to choose the source samples which are most informative for its own classification task. Furthermore, the bilevel nature of the optimization acts as a kind of regularization on the target, mitigating overfitting, as the target does not directly control the representation parameters. Finally, note that the entire process – training of the shared representation, target classifier, and source weights – happens simultaneously.

We pause here to note that the general philosophy behind SOSELETO is related to the literature on *instance reweighting* for domain adaptation, see for example Sugiyama et al. (2008); Yan et al. (2017). However, there is a crucial difference between SOSELETO and this literature, which is related to the difference between domain adaptation and more general transfer learning. Domain adaptation is concerned with the situation in which there is either full overlap between the source and target label sets; or in some more recent work Zhang et al. (2018), partial but significant overlap. Transfer learning, by contrast, refers to the more general situation in which there may be zero overlap between label sets, or possibly very minimal overlap. (For example, if the source consists of natural images and the target of medical images.) The instance reweighting literature is concerned with domain adaptation; the techniques are therefore relevant to the case in which source and target have the same labels. SOSELETO is quite different: it makes no such assumptions, and is therefore a more general approach which can be applied to both "pure" transfer learning, in which there is no overlap between source and target label sets, as well as domain adaptation. (Note also a further distinction with domain adaptation: the target is often – though not always – taken to be unlabelled in domain adaptation. This is not the case for our setting of transfer learning.)

Above, we have illustrated how SOSELETO may be applied to the problem of transfer learning. However, the same algorithm can be applied to the problem of training with noisy labels. Concretely, we assume that there is a large noisy dataset, as well as a much smaller clean dataset; the latter can be constructed cheaply through careful hand-labelling, given its small size. Then if we take the source to be the large noisy dataset, and the target to the small clean dataset, SOSELETO can be applied to the problem. The algorithm will assign high weights to samples with correct labels and

low weights to those with incorrect labels, thereby implicitly denoising the source, and allowing for an accurate classifier to be trained.

The remainder of the paper is organized as follows. Section 2 presents related work. Section 3 presents the SOSELETO algorithm, deriving descent equations as well as convergence properties of the bilevel optimization. Section 4 presents results of experiments on both transfer learning as well as training with noisy labels. Section 5 concludes.

## 2 RELATED WORK

**Transfer learning**  As described in Section 1, the most common techniques for transfer learning are feature extraction and fine-tuning, see for example Donahue et al. (2014) and Girshick et al. (2014), respectively. An older survey of transfer learning techniques may be found in Pan & Yang (2010). Domain adaptation Saenko et al. (2010) is concerned with transferring knowledge when the source and target classes are the same. Earlier techniques aligned the source and target via matching of feature space statistics Tzeng et al. (2014); Long et al. (2015); subsequent work used adversarial methods to improve the domain adaptation performance Ganin & Lempitsky (2015); Tzeng et al. (2015; 2017); Hoffman et al. (2017).

In this paper, we are more interested in transfer learning where the source and target classes are different. A series of recent papers Long et al. (2017); Pei et al. (2018); Cao et al. (2018a;b) address domain adaptation that is closer to our setting. In particular, Cao et al. (2018b) examines "partial transfer learning", the case in which there is partial overlap between source and target classes (particularly when the target classes are a subset of the source). This setting is also dealt with in Busto & Gall (2017).

Ge & Yu (2017) examine the scenario where the source and target classes are completely different. Similar to SOSELETO, they propose selecting a portion of the source dataset. However, the selection is not performed in an end-to-end fashion, as in SOSELETO; rather, selection is performed prior to training, by finding source examples which are similar to the target dataset, where similarity is measured by using filter bank descriptors.

Another recent work of interest is Luo et al. (2017), which focuses on a slightly different scenario: the target consists of a very small number of labelled examples (i.e. the few-shot regime), but a very large number of unlabelled examples. Training is achieved via an adversarial loss to align the source and the target representations, and a special entropy-based loss for the unlabelled part of the data.

**Instance reweighting**  for domain adaptation is a well studied technique, demonstrated e.g. in *Covariate Shift* methods Shimodaira (2000); Sugiyama et al. (2007; 2008). In these works, the source and target label spaces are the same. We, however, allow for different – even entirely non-overlapping – classes in the source and target. Crucially, we do not make assumptions on the similarity of the *distributions* nor do we explicitly optimize for it. The same distinction applies for the recent work of Yan et al. (2017), and for the partial overlap assumption of Zhang et al. (2018). In addition, these two works propose an *unsupervised* approach, whereas our proposed method is completely supervised. Covariate shift determines the weighting for an instance as the ratio of its probability of being in the training set and being in the prediction set. Consequently, the feature vectors are used in re-weighting, regardless of their labels. This renders covariate shift unsuitable for handling noisy labels. Our re-weighing scheme is instead gradient-based and as we show next performs well in this task.

**Learning with noisy labels**  Classification with noisy labels is a longstanding problem in the machine learning literature, see the review paper Frénay & Verleysen (2014) and the references therein. Within the realm of deep learning, it has been observed that with sufficiently large data, learning with label noise – without modification to the learning algorithms – actually leads to reasonably high accuracy Krause et al. (2016); Sun et al. (2017).

The setting that is of greatest interest to us is when the large noisy dataset is accompanied by a small clean dataset. Sukhbaatar et al. (2014) introduce an additional noise layer into the CNN which attempts to adapt the output to align with the noisy label distribution; the parameters of this layer are also learned. Xiao et al. (2015) use a more general noise model, in which the clean label, noisy label, noise type, and image are jointly specified by a probabilistic graphical model. Both the clean

label and the type of noise must be inferred given the image, in this case by two separate CNNs. Li et al. (2017) consider the same setting, but with additional information in the form of a knowledge graph on labels.

Other recent work on label noise includes Rolnick et al. (2017), which shows that adding many copies of an image with noisy labels to a clean dataset barely dents performance; Malach & Shalev-Shwartz (2017), in which two separate networks are simultaneously trained, and a sample only contributes to the gradient descent step if there is disagreement between the networks (if there is agreement, that probably means the label is wrong); and Drory et al. (2018), which analyzes theoretically the situations in which CNNs are more and less resistant to noise. A pair of papers Liu & Tao (2016); Yu et al. (2017) combine ideas of learning with label noise with instance reweighting.

**Bilevel optimization** Bilevel optimization problems have a nested structure: the interior level (sometimes called the lower level) is a standard optimization problem; and the exterior level (sometimes called the upper level) is an optimization problem where the objective is a function of the optimal arguments from the interior level. A branch of mathematical programming, bilevel optimization has been extensively studied within this community Colson et al. (2007); Bard (2013). For recent developments, readers are referred to the review paper Sinha et al. (2018). Bilevel optimization has been used in both machine learning, e.g. Bennett et al. (2006; 2008) and computer vision, e.g. Ochs et al. (2015).

## 3 SOSELETO: SOurce SELEction for Target Optimization

We have two datasets. The source set is the data-rich set, on which we can learn extensively. It is denoted by $\{(x_i^s, y_i^s)\}_{i=1}^{n^s}$, where as usual $x_i^s$ is the $i^{th}$ source training image, and $y_i^s$ is its corresponding label. The second dataset is the target set, which is data-poor; but it is this set which ultimately interests us. That is, the goal in the end is to learn a classifier on the target set, and the source set is only useful insofar as it helps in achieving this goal. The target set is denoted $\{(x_i^t, y_i^t)\}_{i=1}^{n^t}$, and it is assumed that is much smaller than the source set, i.e. $n^t \ll n^s$.

Our goal is to exploit the source set to solve the target classification problem. The key insight is that not all source examples contribute equally useful information in regards to the target problem. For example, suppose that the source set consists of a broad collection of natural images; whereas the target set consists exclusively of various breeds of dog. We would assume that any images of dogs in the source set would help in the target classification task; images of wolves might also help, as might cats. Further afield it might be possible that objects with similar textures as dog fur might be useful, such as rugs. On the flip side, it is probably less likely that images of airplanes and beaches will be relevant (though not impossible). However, the idea is not to come with any preconceived notions (semantic or otherwise) as to which source images will help; rather, the goal is to let the algorithm choose the relevant source images, in an end-to-end fashion.

We assume that the source and target classifier networks have the same architecture, but different network parameters. In particular, the architecture is given by

$$F(x; \theta, \phi)$$

where $\phi$ is last layer, or possibly last few layers, and $\theta$ constitutes all of the remaining layers. We will refer to $\phi$ colloquially as the "classifier", and to $\theta$ as the "features" or "representation". (This is consistent with the usage in related papers, see for example Tzeng et al. (2017).) Now, the source and target will share features, but not classifiers; that is, the source network will be given by $F(x; \theta, \phi^s)$, whereas the target network will be $F(x; \theta, \phi^t)$. The features $\theta$ are shared between the two, and this is what allows for transfer learning.

The *weighted* source loss is given by

$$L_s(\theta, \phi^s, \alpha) = \frac{1}{n^s} \sum_{j=1}^{n^s} \alpha_j \ell(y_j^s, F(x_j^s; \theta, \phi^s))$$

where $\alpha_j \in [0, 1]$ is a weight assigned to each source training example; and $\ell(\cdot, \cdot)$ is a per example classification loss, in this case cross-entropy. The use of the weights $\alpha_j$ will allow us to decide which source images are most relevant for the target classification task.

The target loss is standard:

$$L_t(\theta, \phi^t) = \frac{1}{n^t} \sum_{i=1}^{n^t} \ell(y_i^t, F(x_i^t; \theta, \phi^t))$$

As noted in Section 1, this formulation allows us to address both the transfer learning problem as well as learning with label noise. In the former case, the source and target may have non-overlapping label spaces; high weights will indicate which source examples have relevant knowledge for the target classification task. In the latter case, the source is the noisy dataset, the target is the clean dataset, and they share a classifier (i.e. $\phi^t = \phi^s$) as well as a label space; high weights will indicate which source examples do not have label noise, and are therefore reliable. In either case, the target is much smaller than the source.

The question now becomes: how can we combine the source and target losses into a single optimization problem? A simple idea is to create a weighted sum of source and target losses. Unfortunately, issues are likely to arise regardless of the weight chosen. If the target is weighted equally to the source, then overfitting may likely result given the small size of the target. On the other hand, if the weights are proportional to the size of the two sets, then the source will simply drown out the target.

A more promising idea is to use *bilevel optimization*. Specifically, in the interior level we find the optimal features and source classifier as a function of the weights $\alpha$, by minimizing the source loss:

$$\theta^*(\alpha), \phi^{s*}(\alpha) = \min_{\theta, \phi^s} L_s(\theta, \phi^s, \alpha) \tag{1}$$

In the exterior level, we minimize the target loss, but only through access to the source weights; that is, we solve:

$$\min_{\alpha, \phi^t} L_t(\theta^*(\alpha), \phi^t) \tag{2}$$

Why might we expect this bilevel formulation to succeed? The key is that the target only has access to the features in an *indirect* manner, by controlling which source examples are included in the source classification problem. Thus, the target can influence the features chosen, but only in this roundabout way. This serves as an extra form of regularization, mitigating overfitting, which is the main threat when dealing with a small set such as the target.

Implementing the bilevel optimization is rendered somewhat challenging due to the need to solve the optimization problem in the interior level (1). Note that this optimization problem must be solved at every point in time; thus, if we choose to solve the optimization (2) for the exterior level via gradient descent, we will need to solve the interior level optimization (1) at each iteration of the gradient descent. This is clearly inefficient. Furthermore, it is counter to the standard deep learning practice of taking small steps which improve the loss. Thus, we instead propose the following procedure.

At a given iteration, we will take a gradient descent step for the interior level problem (1):

$$\begin{aligned}
\theta_{m+1} &= \theta_m - \lambda_p \frac{\partial L_s}{\partial \theta}(\theta_m, \phi_m^s, \alpha_m) \\
&= \theta_m - \lambda_p Q(\theta_m, \phi_m^s)\alpha_m
\end{aligned} \tag{3}$$

where $m$ is the iteration number; $\lambda_p$ is the learning rate (where the subscript $p$ stands for "parameters", to distinguish it from a second learning rate for $\alpha$, to appear shortly); and $Q(\theta, \phi^s)$ is a matrix whose $j^{th}$ column is given by

$$q_j = \frac{1}{n^s} \frac{\partial}{\partial \theta} \ell(y_j^s, F(x_j^s; \theta, \phi^s))$$

Thus, Equation (3) leads to an improvement in the features $\theta$, for a fixed set of source weights $\alpha$. Note that there will be an identical descent equation for the classifier $\phi^s$, which we omit for clarity.

Given this iterative version of the interior level of the bilevel optimization, we may now turn to the exterior level. Plugging Equation (3) into Equation (2) gives the following problem:

$$\min_{\alpha, \phi^t} L_t(\theta_m - \lambda_p Q\alpha, \phi^t)$$

---

**Algorithm 1** SOSELETO: SOurce SELEction for Target Optimization

Initialize: $\theta$, $\phi^s$, $\alpha$, $\phi^t$.
**while** not converged **do**
    Sample source batch $b \leftarrow \{b_1, \ldots, b_L\} \subset \{1, \ldots, n^s\}$
    Denote by $\alpha_b = [\alpha_{b_1}, \ldots, \alpha_{b_L}]$
    $Q \leftarrow [q_1 \ldots q_L]$   where   $q_\ell \leftarrow \frac{1}{n^s} \frac{\partial}{\partial \theta} \ell(y_{b_\ell}^s, F(x_{b_\ell}^s; \theta, \phi^s))$
    $R \leftarrow [r_1 \ldots r_L]$   where   $r_\ell \leftarrow \frac{1}{n^s} \frac{\partial}{\partial \phi^s} \ell(y_{b_\ell}^s, F(x_{b_\ell}^s; \theta, \phi^s))$
    $\theta \leftarrow \theta - \lambda_p Q \alpha_b$
    $\phi^s \leftarrow \phi^s - \lambda_p R \alpha_b$
    $\alpha_b \leftarrow \text{CLIP}_{[0,1]} \left( \alpha_b + \lambda_\alpha \lambda_p Q^T \frac{\partial L_t}{\partial \theta} \right)$
    $\phi^t \leftarrow \phi^t - \lambda_p \frac{\partial L_t}{\partial \phi^t}$
**end while**

---

where we have suppressed $Q$'s arguments for readability. We can then take a gradient descent step of this equation, yielding:

$$\alpha_{m+1} = \alpha_m - \lambda_\alpha \frac{\partial}{\partial \alpha} L_t(\theta_m - \lambda_p Q\alpha, \phi^t)$$

$$= \alpha_m + \lambda_\alpha \lambda_p Q^T \frac{\partial L_t}{\partial \theta}(\theta_m - Q\alpha_m \lambda_p)$$

$$\approx \alpha_m + \lambda_\alpha \lambda_p Q^T \frac{\partial L_t}{\partial \theta}(\theta_m) \tag{4}$$

where in the final line, we have made use of the fact that $\lambda_p$ is small. Of course, there will also be a descent equation for the classifier $\phi^t$. The resulting update scheme is quite intuitive: source example weights are update according to how well they align with the target aggregated gradient.

We have not yet dealt with the weight constraint. That is, we would like to explicitly require that each $\alpha_j \in [0, 1]$. We may achieve this by requiring $\alpha_j = \sigma(\beta_j)$ where the new variable $\beta_j \in \mathbb{R}$, and $\sigma : \mathbb{R} \to [0, 1]$ is a sigmoid-type function. As shown in Appendix A, for a particular piecewise linear sigmoid function, replacing the Update Equation (4) with a corresponding update equation for $\beta$ is equivalent to modifying Equation (4) to read

$$\alpha_{m+1} = \text{CLIP}_{[0,1]} \left( \alpha_m + \lambda_\alpha \lambda_p Q^T \frac{\partial L_t}{\partial \theta}(\theta_m) \right) \tag{5}$$

where $\text{CLIP}_{[0,1]}$ clips the values below $0$ to be $0$; and above $1$ to be $1$.

Thus, SOSELETO consists of alternating Equations (3) and (5), along with the descent equations for the source and target classifiers $\phi^s$ and $\phi^t$. As usual, the whole operation is done on a mini-batch basis, rather than using the entire set; note that if processing is done in parallel, then source mini-batches are taken to be non-overlapping, so as to avoid conflicts in the weight updates. SOSELETO is summarized in Algorithm 1. Note that the target derivatives $\partial L_t / \partial \theta$ and $\partial L_t / \partial \phi^t$ are evaluated over a target mini-batch; we suppress this for clarity.

In terms of time-complexity, we note that each iteration requires both a source batch and a target batch; assuming identical batch sizes, this means that SOSELETO requires about twice the time as the ordinary source classification problem. Regarding space-complexity, in addition to the ordinary network parameters we need to store the source weights $\alpha$. Thus, the additional relative space-complexity required is the ratio of the source dataset size to the number of network parameters. This is obviously problem and architecture dependent; a typical number might be given by taking the source dataset to be Imagenet ILSVRC-2012 (size 1.2M) and the architecture to be ResNeXt-101 Xie et al. (2017) (size 44.3M parameters), yielding a relative space increase of about 3%.

**Convergence properties**  SOSELETO is only an approximation to the solution of a bilevel optimization problem. As a result, it is not entirely clear whether it will even converge. In Appendix B, we demonstrate a set of sufficient conditions for SOSELETO to converge to a local minimum of the target loss $L_t$.

## 4 RESULTS

We briefly discuss some implementation details. In all experiments, we use the SGD optimizer without learning rate decay, and we use $\lambda_\alpha = 1$. We initialize the $\alpha$-values to be 1, and in practice clip them to be in the slightly expanded range $[0, 1.1]$; this allows more relevant source points some room to grow. Other settings are experiment specific, and are discussed in the relevant sections.

### 4.1 NOISY LABELS: SYNTHETIC EXPERIMENT

To illustrate how SOSELETO works on the problem of learning with noisy labels, we begin with a synthetic experiment, see Figure 1. The setting is straightforward: the source dataset consists of 500 points which lie in $\mathbb{R}^2$. There are two labels / classes, and the ideal separator between the classes is the $y$-axis. However, of the 500 points, 100 are corrupted: that is, they lie on the wrong side of the separator. This is shown in Figure 1(a), in which one class is shown as white triangles and the second as black pluses. The target dataset is a set of 50 points, which are "clean", in the sense that they lie on the correct sides of the separator. (For the sake of simplicity, the target set is not illustrated.)

SOSELETO is run for 100 epochs. In Figures 1(b) and 1(c), we choose a threshold of 0.1 on the weights $\alpha$, and colour the points accordingly. In particular, in Figure 1(b) the clean (i.e. correctly labelled) instances which are above the threshold are labelled in green, while those below the threshold are labelled in red; as can be seen, all of the clean points lie above the threshold for this choice of threshold, meaning that SOSELETO has correctly identified all of the clean points. In Figure 1(c), the noisy (i.e. incorrectly labelled) instances which are *below* the threshold are labelled in green; and those above the threshold are labelled in red. In this case, SOSELETO correctly identifies most of these noisy labels by assigning them small weights (below 0.1); in fact, 92 out of 100 points are assigned such small weights. The remaining 8 points, those shown in red, are all near the separator, and it is therefore not very surprising that SOSELETO mislabels them. All told, using this particular threshold the algorithm correctly accounts for 492 out of 500 points, i.e. 98.4%.

Further analysis appears in Figures 1(d) and 1(e). In Figure 1(e), a plot is shown of mean weight vs. training epoch for clean instances and noisy instances; the width of each plot is the 95% confidence interval of the weights of that type. All weights are initialized at 0.5; after 100 epochs, the clean instances have a mean weight of about 0.8, whereas the noisy instances have a mean weight of about 0.05. The evolution is exactly as one would expect. Figure 1(e) examines the role of the threshold, chose as 0.1 in the above discussion; although 0.1 is a good choice in this case, the good behaviour is fairly robust to choices in the range of 0.1 to 0.4.

### 4.2 NOISY LABELS: CIFAR-10

We now turn to a real-world setting of the problem of learning with label noise. We use a noisy version of CIFAR-10 Krizhevsky & Hinton (2009), following the settings used in Sukhbaatar et al. (2014); Xiao et al. (2015). In particular, an overall noise level is selected. Based on this, a label confusion matrix is chosen such that the diagonal entries of the matrix are equal to one minus the noise level, and the off-diagonals are chosen randomly (while maintaining the matrix's stochasticity). Noisy labels are then sampled according to this confusion matrix. We run experiments for various overall noise levels.

The target consists of a small clean dataset. CIFAR-10's train set consists of 50K images; of this 50K, both Sukhbaatar et al. (2014); Xiao et al. (2015) set aside 10K clean examples for pre-training, a necessary step in both of these algorithms. In contrast, we use a smaller clean dataset of half the size, i.e. 5K examples while the rest of the 45K samples are noisy. We compare our results to the two state of the art methods Sukhbaatar et al. (2014); Xiao et al. (2015), as they both address the same setting as we do – the large noisy dataset is accompanied by a small clean dataset, with no extra side-information available. In addition, we compare with the baseline of simply training on the noisy labels without modification. In all cases, Caffes CIFAR-10 Quick cif architecture has been used. For SOSELETO, we use the following settings: $\lambda_p = 10^{-4}$, the target batch-size is 32, and the source batch-size is 256. We use a larger source batch-size to enable more $\alpha$-values to be affected quickly.

Table 1: Noisy labels: CIFAR-10. Best results in bold.

| Noise Level | CIFAR-10 Quick – | Sukhbaatar et al. (2014) 10K clean examples | Xiao et al. (2015) 10K clean examples | SOSELETO 5K clean examples |
|---|---|---|---|---|
| 30% | 65.57 | 69.73 | 69.81 | **72.41** |
| 40% | 62.38 | 66.66 | 66.76 | **69.98** |
| 50% | 57.36 | 63.39 | 63.00 | **66.33** |

Table 2: SVHN 0-4 $\rightarrow$ MNIST 5-9. Best results in bold.

| Uses Unlabelled Data? | Method | $n^t = 20$ | $n^t = 25$ |
|---|---|---|---|
| No | Target only | 80.1 | 84.0 |
| No | Fine-tuning | 80.2 | 83.0 |
| No | SOSELETO | **83.2** | **87.9** |
| Yes | Vinyals et al. (2016) | 56.6 | 51.3 |
| Yes | Fine-tuned variant of Vinyals et al. (2016) | 79.3 | 82.7 |
| Yes | Luo et al. (2017) | 80.4 | 83.1 |
| Yes | Label-efficient version of Luo et al. (2017) | **94.2** | **95.0** |

Results are shown in Table 1 for three different overall noise levels, 30%, 40%, and 50%. Performance is reported for CIFAR-10's test set, which is of size 10K. (Note that the competitors' performance numbers are taken from Xiao et al. (2015).) SOSELETO achieves state of the art on all three noise levels, with considerably better performance than both Sukhbaatar et al. (2014) and Xiao et al. (2015): between 2.6% to 3.2% absolute improvement. Furthermore, it does so in each case with only half of the clean samples used in Sukhbaatar et al. (2014); Xiao et al. (2015).

We perform further analysis by examining the $\alpha$-values that SOSELETO chooses on convergence, see Figure 4.2. To visualize the results, we imagine thresholding the training samples in the source set on the basis of their $\alpha$-values; we only keep those samples with $\alpha$ greater than a given threshold. By increasing the threshold, we both reduce the total number of samples available, as well as change the effective noise level, which is the fraction of remaining samples which have incorrect labels. We may therefore plot these two quantities against each other, as shown in Figure 4.2; we show three plots, one for each noise level. Looking at these plots, we see for example that for the 30% noise level, if we take the half of the training samples with the highest $\alpha$-values, we are left with only about 4% which have incorrect labels. We can therefore see that SOSELETO has effectively filtered out the incorrect labels in this instance. For the 40% and 50% noise levels, the corresponding numbers are about 10% and 20% incorrect labels; while not as effective in the 30% noise level, SOSELETO is still operating as designed. Further evidence for this is provided by the large slopes of all three curves on the righthand side of the graph.

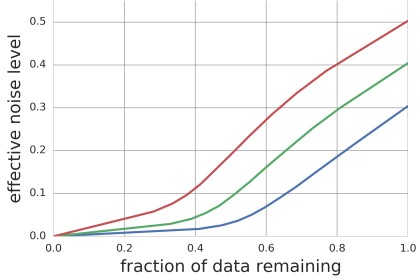

Figure 2: Noisy labels on CIFAR-10: Effect of $\alpha$-values chosen by SOSELETO. Blue is 30% noise, green is 40% noise, red is 50% noise. See accompanying explanation in the text.

### 4.3 TRANSFER LEARNING: SVHN 0-4 TO MNIST 5-9

We now examine the performance of SOSELETO on a transfer learning task. In order to provide a challenging setting, we choose to (a) use source and target sets with disjoint label sets, and (b) use a very small target set. In particular, the source dataset is chosen to the subset of Google Street View House Numbers (SVHN) Netzer et al. (2011) corresponding to digits 0-4. SVHN's train set is of size 73,257 images, with about half of those belonging to the digits 0-4. The target dataset is a very small subset of MNIST LeCun et al. (1998) corresponding to digits 5-9. While MNIST's train set is of size 60K, with 30K corresponding to digits 5-9, we use very small subsets: either 20 or 25 images, with equal numbers sampled from each class (4 and 5, respectively). Thus, as mentioned, there is no overlap between source and target classes, making it a true transfer learning (rather than domain adaptation) problem; and the small target set size adds further challenge. Furthermore, this task has already been examined in Luo et al. (2017).

We compare our results with the following techniques. Target only, which indicates training on just the target set; standard fine-tuning; Matching Nets Vinyals et al. (2016), a few-shot technique which is relevant given the small target size; fine-tuned Matching Nets, in which the previous result is then fine-tuned on the target set; and two variants of the Label Efficient Learning technique Luo et al. (2017) – one which includes fine-tuning plus a domain adversarial loss, and the other the full technique presented in Luo et al. (2017). Note that besides the target only and fine-tuning approaches, all other approaches *depend on unlabelled target data*. Specifically, they use all of the remaining MNIST 5-9 examples – about 30,000 – in order to aid in transfer learning. SOSELETO, by contrast, does not make use of any of this data.

For each of the above methods, the simple LeNet architecture LeCun et al. (1998) was used. For SOSELETO, we use the following settings: $\lambda_p = 10^{-2}$, the source batch-size is 32, and the target batch-size is 10 (it is chosen to be small since the target itself is very small). Additionally, the SVHN images were resized to $28 \times 28$, to match the MNIST size. The performance of the various methods is shown in Table 2, and is reported for MNIST's test set which is of size 10K. We have divided Table 2 into two parts: those techniques which use the 30K examples of unlabelled data, and those which do not. SOSELETO has superior performance to all of the techniques which do not use unlabelled data. Furthermore, SOSELETO has superior performance to all of the techniques which *do* use unlabelled data, except the Label Efficient technique. It is noteworthy in particular that SOSELETO outperforms the few-shot techniques, despite not being designed to deal with such small amounts of data.

In Appendix C we further analyze which SVHN instances are considered more useful than others by SOSELETO, by transfering all of SVHN classes to MNSIT 5-9.

**Two-stage SOSELETO** Finally, we note that although SOSELETO is not designed to use unlabelled data, one may do so using the following two-stage procedure. Stage 1: run SOSELETO as described above. Stage 2: use the learned SOSELETO classifier to classify the unlabelled data. This will now constitute a dataset with noisy labels, and SOSELETO can now be run in the mode of training with label noise, where the noisily labelled unsupervised data is now the source, and the target remains the same small clean set. In the case of $n^t = 25$, this procedure elevates the accuracy to above **92**%.

## 5 CONCLUSIONS

We have presented SOSELETO, a technique for exploiting a source dataset to learn a target classification task. This exploitation takes the form of joint training through bilevel optimization, in which the source loss is weighted by sample, and is optimized with respect to the network parameters; while the target loss is optimized with respect to these weights and its own classifier. We have derived an efficient algorithm for performing this bilevel optimization, through joint descent in the network parameters and the source weights, and have analyzed the algorithm's convergence properties. We have empirically shown the effectiveness of the algorithm on both learning with label noise, as well as transfer learning problems. An interesting direction for future research involves incorporating an additional domain alignment term into SOSELETO, in the case where the source and target dataset have overlapping labels. We note that SOSELETO is architecture-agnostic, and thus may be easily deployed. Furthermore, although we have focused on classification tasks, the

technique is general and may be applied to other learning tasks within computer vision; this is an important direction for future research.

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

## APPENDIX A    CONSTRAINING THE WEIGHTS

Recall that our goal is to explicitly require that $\alpha_j \in [0, 1]$. We may achieve this by requiring

$$\alpha_j = \sigma(\beta_j) = \begin{cases} 0 & \text{if } \beta_j < 0 \\ \beta_j & \text{if } 0 \leq \beta_j \leq 1 \\ 1 & \text{if } \beta_j > 1 \end{cases}$$

where the new variable $\beta_j \in \mathbb{R}$, and $\sigma(\cdot)$ is a kind of piecewise linear sigmoid function.

Now we will wish to replace the Update Equation (4), the update for $\alpha$, with a corresponding update equation for $\beta$. This is straightforward. Define the Jacobian $\partial\alpha/\partial\beta$ by

$$\left(\frac{\partial\alpha}{\partial\beta}\right)_{ij} = \frac{\partial\alpha_i}{\partial\beta_j}$$

Then we modify Equation (4) to read

$$\beta_{m+1} = \beta_m + \lambda_\alpha\lambda_p \left(\frac{\partial\alpha}{\partial\beta}\right)^T Q^T \frac{\partial L_t}{\partial\theta}(\theta_m)$$

The Jacobian is easy to compute analytically:

$$\frac{\partial\alpha}{\partial\beta} = \text{diag}(\sigma'(\beta_j)), \quad \text{where} \quad \sigma'(z) = \begin{cases} 0 & \text{if } z < 0 \\ 1 & \text{if } 0 \leq z \leq 1 \\ 0 & \text{if } z > 1 \end{cases}$$

Due to this very simple form, it is easy to see that $\beta_m$ will never lie outside of $[0, 1]$; and thus that $\alpha_m = \beta_m$ for all time. Hence, we can simply replace this equation with

$$\alpha_{m+1} = \text{CLIP}_{[0,1]} \left(\alpha_m + \lambda_\alpha\lambda_p Q^T \frac{\partial L_t}{\partial\theta}(\theta_m)\right)$$

where $\text{CLIP}_{[0,1]}$ clips the values below 0 to be 0; and above 1 to be 1.

## APPENDIX B    PROOF OF CONVERGENCE

SOWETO is only an approximation to the solution of a bilevel optimization problem. As a result, it is not entirely clear whether it will even converge. In this section, we demonstrate a set of sufficient conditions for SOWETO to converge to a local minimum of the target loss $L_t$.

To this end, let us examine the change in the target loss from iteration $m$ to $m + 1$:

$$\Delta L_t = L_t(\theta_{m+1}, \phi^t_{m+1}) - L_t(\theta_m, \phi^t_m)$$

$$= L_t \left(\theta_m - \lambda_p Q\alpha_m, \; \phi^t_m - \lambda_p \frac{\partial L_t}{\partial\phi^t}\right) - L_t(\theta_m, \phi^t_m)$$

$$\approx L_t(\theta_m, \phi^t_m) - \lambda_p \left(\frac{\partial L_t}{\partial\theta}\right)^T Q\alpha_m - \lambda_p \left(\frac{\partial L_t}{\partial\phi^t}\right)^T \frac{\partial L_t}{\partial\phi^t} - L_t(\theta_m, \phi^t_m)$$

$$= -\lambda_p \left(\frac{\partial L_t}{\partial\theta}\right)^T Q\alpha_m - \lambda_p \left\|\frac{\partial L_t}{\partial\phi^t}\right\|^2$$

Now, we can use the evolution of the weights $\alpha$. Specifically, we substitute Equation (4) into the above, to get

$$\Delta L_t \approx -\lambda_p \left(\frac{\partial L_t}{\partial\theta}\right)^T Q \left(\alpha_{m-1} + \lambda_\alpha\lambda_p Q^T \frac{\partial L_t}{\partial\theta}\right) - \lambda_p \left\|\frac{\partial L_t}{\partial\phi^t}\right\|^2$$

$$= -\lambda_p \left(\frac{\partial L_t}{\partial\theta}\right)^T Q\alpha_{m-1} - \lambda_\alpha\lambda_p^2 \left\|Q^T \frac{\partial L_t}{\partial\theta}\right\|^2 - \lambda_p \left\|\frac{\partial L_t}{\partial\phi^t}\right\|^2$$

$$\equiv \Delta L_t^{FO}$$

where $\Delta L_t^{FO}$ indicates the change in the target loss, to first order.

Note that the latter two terms in $\Delta L_t^{FO}$ are both negative, and will therefore cause the first order approximation of the target loss to decrease, as desired. As regards the first term, matters are unclear. However, it is clear that if we set the learning rate $\lambda_\alpha$ sufficiently large, the second term will eventually dominate the first term, and the target loss will be decreased. Indeed, we can do a slightly finer analysis. Ignoring the final term (which is always negative), and setting $v = Q^T \frac{\partial L_t}{\partial \theta}$, we have that

$$\begin{aligned}
\Delta L_t^{FO} &\leq -\lambda_p v^T \alpha_{m-1} - \lambda_\alpha \lambda_p^2 \|v\|^2 \\
&\leq \lambda_p \|v\|_1 - \lambda_\alpha \lambda_p^2 \|v\|_2^2 \\
&\leq \lambda_p \sqrt{n^s} \|v\|_2 - \lambda_\alpha \lambda_p^2 \|v\|_2^2 \\
&= \lambda_p \|v\|_2 \left( \sqrt{n^s} - \lambda_\alpha \lambda_p \|v\|_2 \right)
\end{aligned}$$

where in the second line we have used the fact that all elements of $\alpha$ are in $[0, 1]$; and in the third line, we have used a standard bound on the $L_1$ norm of a vector.

Thus, a sufficient condition for the first order approximation of the target loss to decrease is if

$$\lambda_\alpha \geq \frac{\sqrt{n^s}}{\lambda_p \left\| Q^T \frac{\partial L_t}{\partial \theta} \right\|}$$

If this is true at all iterations, then the target loss will continually decrease and converge to a local minimum (given that the loss is bounded from below by $0$).

## APPENDIX C    ANALYZING SVHN 0-9 TO MNIST 5-9

SOSELETO is capable of automatically pruning unhelpful instances at train time. The experiment presented in Section 4.3 demonstrates how SOSELETO can improve classification of MNIST 5-9 by making use of different digits from a different dataset (SVHN 0-4). To further reason about which instances are chosen as useful, we have conducted another experiment: SVHN 0-9 to MNIST 5-9. There is now a partial overlap in classes between source and target. Our findings are summarized in what follows. An immediate effect of increasing the source set, was a dramatic improvement in accuracy to $90.3\%$.

Measuring the percentage of "good" instances (i.e. instances with weight above a certain threshold) didn't reveal a strong correlation with the labels. In Figure 3 we show this result for a threshold of $0.8$. As can be seen, labels 7-9 are slightly higher than the rest but there is no strong evidence of labels 5-9 being more useful than 0-4, as one might hope for.

That said, a more careful examination of low- and high-weighted instances, revealed that the usefulness of an instance, as determined by SOSELETO, is more tied to its appearance: namely, whether the digit is centered, at a similar size as MNIST, the amount of blur, and rotation. In Figure 4 we show a random sample of some "good" and "bad" (i.e. high and low weights, respectively). A close look reveals that "good" instances often tend to be complete, centered, axis aligned, and at a good size (wrt MNIST sizes). Especially interesting was that, among the "bad" instances, we found about $3 - 5\%$ wrongly labeled instances! In Figure 5 we display several especially problematic instances of the SVHN, all of which are labeled as "0" in the dataset. As can be seen, some examples are very clear errors. The highly weighted instances, on the other hand, had almost no such errors.

% instances above threshold

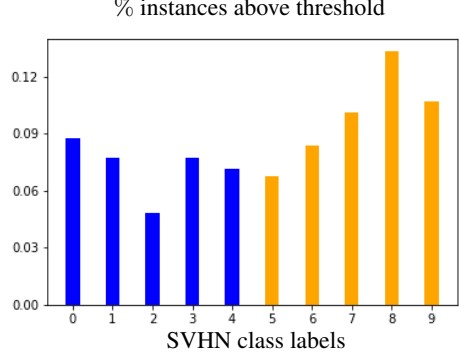

SVHN class labels

Figure 3: Percentage of good instances from SVHN per class. Classes 0-4 are colored blue and classes 5-9 are colored orange.

Randomly sampled "good" instances     Randomly sampled "bad" instances

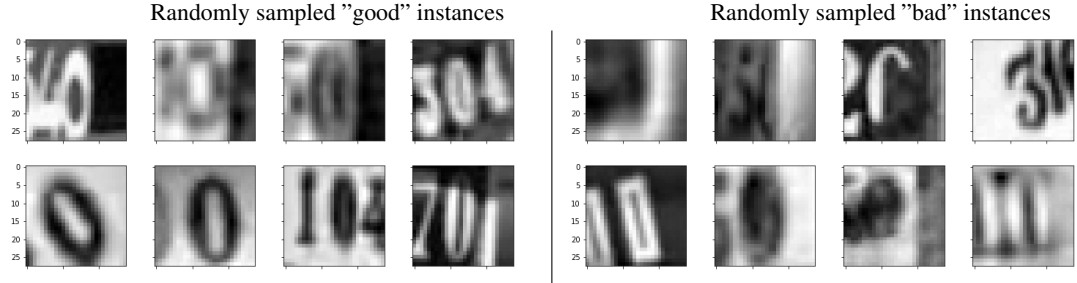

Figure 4: SVHN "good" (left) and "bad" (right) instances of class label 0.

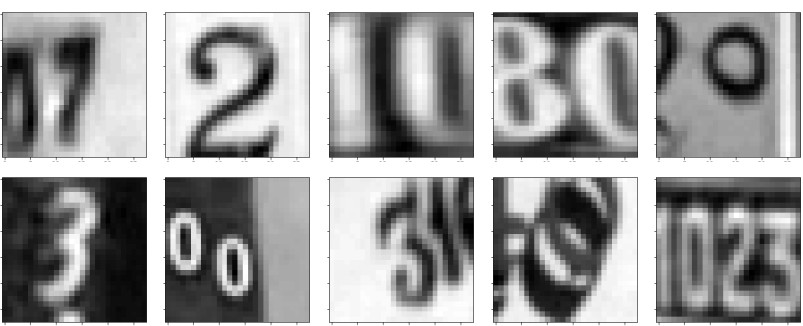

Figure 5: Hand-picked examples from the pool of "bad" instances with label 0.

