# OpenReview forum: "SOSELETO: A Unified Approach to Transfer Learning and Training with Noisy Labels"
_ICLR.cc/2019/Conference_

### Official Review · AnonReviewer3 · 2018-11-01

**Rating:** 5
**Confidence:** 4

**Review:**

In this paper, the authors propose a SOSELETO (source selection for target optimization) framework to transfer learning and training with noisy labels. The intuition is some source instances are more informative than the others. Specifically, source instances are weighted and the weights are learned in a bilevel optimization scheme. Experimental studies on both training with noisy label problems and transfer learning problems demonstrate the effectiveness of the proposed SOSELETO.

Overall, this paper is well-written, and easy to follow. The intuition is clear and reasonable, although it is not new. Regarding the technical section, I have the following comments:
(1)	The paper assumes that the source and target domains share the same feature representation parameters \theta. This is a widely used assumption in the existing works. However, these works usually have a specific part to align two domains to support the assumption, e.g. adversarial loss or MMD. In objective of SOSELETO, I do not see such a domain alignment part. I am wondering whether the assumption is still valid in this case. From the experimental study, I find SOSELETO achieves very good results in transfer learning problems. I am wondering whether the performance would be further improved if a domain alignment objective is added in the weighted source loss.
(2)	Each source has a weight, and thus there are n^s \alpha. As mini-batch is used in the training, I am wondering whether batches are overlapping or not. If overlapping, how to decide the final \alpha_i for x^s_i as you may obtain several \alpha_i in batches.
(3)	Another point is abouth \lambda_p. In the contents, you omit the last term Q \alpha_m \lambda_p in eq.(4) as you use the fact that it is very small. I am not convincing on this omission as \lambda_p is also a weight for the entire derivative. Moreover, if \lambda_p is very small, the convergence would be very slow. In the experimental studies, you use different \lambda_p for different problems. Then, what’s the rule of setting \lambda_p given a new problem?

Regarding the experimental results, the experimental settings for the section 4.2 are not very clear to me. You may need to clearly state the train and test set (e.g. data size) for each method.

---

> ### Author Response · Authors · 2018-11-23
> **Addressing particular concerns raised by R3**
>
> Comment: The paper assumes that the source and target domains share the same feature representation parameters \theta. This is a widely used assumption in the existing works. However, these works usually have a specific part to align two domains to support the assumption, e.g. adversarial loss or MMD.  In objective of SOSELETO, I do not see such a domain alignment part. I am wondering whether the assumption is still valid in this case. From the experimental study, I find SOSELETO achieves very good results in transfer learning problems. I am wondering whether the performance would be further improved if a domain alignment objective is added in the weighted source loss.
> Response: This is a very interesting point.  The main reason we do not use domain alignment is that this technique is most applicable to the case of domain adapation, where the source and target label sets are overlapping (or at least largely overlapping); and less relevant in the case of transfer learning, where there may be no overlap between source and target label set.  However, it might be interesting to try out the domain alignment as an additional term in cases where the label sets are overlapping, e.g. in the case of noisy labels.
> We have added extra text in Section 5 pointing this out as interesting direction for future research.
>
>
> Comment: Each source has a weight, and thus there are n^s \alpha. As mini-batch is used in the training, I am wondering whether batches are overlapping or not. If overlapping, how to decide the final \alpha_i for x^s_i as you may obtain several \alpha_i in batches.
> Response: The batches used are non-overlapping, as you have correctly guessed.  A sentence has been added to Section 3 to clarify this point.
>
>
> Comment: Another point is about \lambda_p. In the contents, you omit the last term Q \alpha_m \lambda_p in eq.(4) as you use the fact that it is very small. I am not convincing on this omission as \lambda_p is also a weight for the entire derivative. Moreover, if \lambda_p is very small, the convergence would be very slow. In the experimental studies, you use different \lambda_p for different problems. Then, what’s the rule of setting \lambda_p given a new problem?
> Response: What we have done is a standard first-order approximation.  In practice, \lamda_p is taken very small: either 1e-4 or 1e-2, depending on the application.  Note that this is as opposed to \lambda_\alpha, which is taken to be considerably larger (in practice, 1).
> Regarding the rule for choosing \lambda_p: \lambda_p plays much the same rule as the standard learning rate in deep learning, so it varies by problem.  As with the leraning rate, one chooses a value which leads to a steady decrease in the loss.  Again, note that \lambda_\alpha does not vary by problem, and is always fixed at 1.
>
> Comment: Regarding the experimental results, the experimental settings for the section 4.2 are not very clear to me. You may need to clearly state the train and test set (e.g. data size) for each method.
> Response: Both competing methods split the 50K train-set to a 10K clean samples and and 40K noisy samples. We use just 5K clean samples and 45K noisy samples. So, strictly less clean data is needed to achive better accuracy. We have included this in detail in the modified menuscript.

---

### Official Review · AnonReviewer2 · 2018-11-03
**Interpretation needed for the weights**

**Rating:** 5
**Confidence:** 4

**Review:**

This is an interesting paper claiming that on assumptions are made (or explicitly made) on the similarity of distributions. Traditionally, we learned the weights for transfer learning by matching the distributions. I am wondering if there are any relationships between those two methods. It is necessary to show the differences between the weighted source domain and the target domain, and compare them with the traditional matching methods.

My another concern is about the technical contribution. The model is very intuitive and simple. Some analyses are made for optimization. However, theoretical justifications are lacking, making the technical contribution weak and looks like a simple combination of two existing techniques. I would like to know if the weights are identifiable and what kinds of weights are preferred.

By searching, I found related papers on transfer learning with label noise and learning with label noise by importance reweighting, e.g., Yu, Xiyu, et al. "Transfer Learning with Label Noise." arXiv preprint arXiv:1707.09724 (2017). and Liu, Tongliang, and Dacheng Tao. "Classification with noisy labels by importance reweighting." IEEE Transactions on pattern analysis and machine intelligence 38.3 (2016): 447-461. However, they are not discussed in the submission. It is curious to see the relationships and differences.

---

> ### Author Response · Authors · 2018-11-23
> **Addressing particular concerns raised by R2**
>
> Comment: This is an interesting paper claiming that on assumptions are made (or explicitly made) on the similarity of distributions. Traditionally, we learned the weights for transfer learning by matching the distributions. I am wondering if there are any relationships between those two methods. It is necessary to show the differences between the weighted source domain and the target domain, and compare them with the traditional matching methods.
> Response: Please see “General Response to All Reviewers”.  The traditional reweighting methods do not apply to the case of Transfer Learning (= non-overlapping source and target label sets), and would therefore not be relevant to - for example - the SVHN 0-4 to MNIST 5-9 experiment.  Our technique is more general, and therefore we compare with other techniques with the same degree of generality.
>
> Comment: My another concern is about the technical contribution. The model is very intuitive and simple. Some analyses are made for optimization. However, theoretical justifications are lacking.
> Response: Respectfully, we believe that intuitiveness and simplicity are virtues.  The simplicity means that the method can be plugged into many different learning techniques, as we discuss.
>
> Comment: I would like to know if the weights are identifiable and what kinds of weights are preferred.
> Response: Please see “General Response to All Reviewers”.
>
> Comment: By searching, I found related papers on transfer learning with label noise and learning with label noise by importance reweighting, e.g., Yu, Xiyu, et al. "Transfer Learning with Label Noise." arXiv preprint arXiv:1707.09724 (2017). and Liu, Tongliang, and Dacheng Tao. "Classification with noisy labels by importance reweighting." IEEE Transactions on pattern analysis and machine intelligence 38.3 (2016): 447-461. However, they are not discussed in the submission. It is curious to see the relationships and differences.
> Response: Please see “General Response to All Reviewers”.  These new references have been incorporated in Section 2, Related Work.

---

### Official Review · AnonReviewer1 · 2018-11-05
**Interesting approach to transfer learning, although the experimental section could have been clearer**

**Rating:** 7
**Confidence:** 4

**Review:**

PROS:
* This is an interesting approach of assigning contribution weights to each source sample.
* Could be very helpful for tasks where we have a noisy and a (small) clean dataset.
* The method seems to be performing well for the tasks chosen, especially for the CIFAR experiments.
* Simple idea and relatively easy to implement

CONS:
* Clarity could be improved, especially in the experimental section
* The motivation for the SVHN 0-4 to MNIST 5-9 is not clear. It would make more sense to me to transfer between SVHN 0-5 to SVHN 5-9, or from the entire SVHN to the entire MNIST, but this particular transfer seems somewhat irrelevant to the claims. The two domains are particularly dissimilar and trying to select "good" SVHN samples according to 20 or 25 MNIST samples seems somewhat ill-posed. It is also particularly surprising to me that 25 MNIST samples were enough to train a LeNet to the point of 84% accuracy on the entire MNIST test set. (I'm referring to the target-only line) Is that really the case, or was a larger training set used for that particular line?
* There is a claim that "SOSELETO has superior performance to all of the techniqiues which do not use unlabelled data", however I'm not sure whether these techniques were used as prescribed and if the comparison was fair. For example, I believe domain adaptation techniques like DANN, largely assume a common label space between the domains.
* Comparison with previous re-weighting techniques would have been very informative.

QUALITY:
* The quality of the writing was overall high, with a few exceptions, including the related work and the experimental section.
* In related work, the "bilevel optimization" section could be a bit more descriptive, maybe some of the explanationgiven in Sect. 3 could be moved here?
* The experiments were convincing, with the exception of the SVHN to MNIST section.

CLARITY:
* I believe a better synthetic experiment could be chosen to highlight the approach: how about a truly noisy dataset that is not as separable as the "noisy" dataset in Figure 1? Maybe you could have the same noisy dataset but with a small portion of random points having the wrong label. For the same experiment, it should be clearly stated that your task is binary classification and what was the classifier used.
* For the CIFAR experiments, it is very good that it performs well, but it'd be informative to see if SOSELETO can perform even better with 10K samples.
* It wasn't clear to me whether the a-values of only one batch (32 samples?) at a time were affected. If so, how does this scale to really large datasets like, say, Imagenet?
* In the CIFAR experiments, it is mentioned that a target batch-size is chosen to be larger to enable more a-values to be affected. This seems like a typo, but it was confusing. (I assume that the source batch-size is chosen to be larger)
* Figure 2 could use a better caption and a legend. It would also be an easier figure to parse if the x-axis was reversed (eg. if the x-axis was the fraction of data used)
* It was not clear to me what "true transfer learning" means as opposed to domain adaptation.

ORIGINALITY:
* It seems that this idea has been explored before, however I'm not personally familiar with that work. I would have definitely liked to see comparisons with it though.


SIGNIFICANCE:
* This is a simple idea that seems to work well. As I wrote above, it would be great to know how it compares to other re-weighting techniques.

---

> ### Author Response · Authors · 2018-11-23
> **Addressing particular concerns raised by R1**
>
> Comment: Clarity could be improved, especially in the experimental section.
> Response: We have accounted for this through the inclusion of the two new experiments, see “General Response to All Reviewers”.
>
> Comment: The motivation for the SVHN 0-4 to MNIST 5-9 is not clear. It would make more sense to me to transfer between SVHN 0-5 to SVHN 5-9, or from the entire SVHN to the entire MNIST, but this particular transfer seems somewhat irrelevant to the claims.
> Response: Please see “General Response to All Reviewers”.  As SOSELETO is capable of automatically pruning useful instances at train time, the original experiment with zero-overlap in class labels was meant to demonstrate an extreme version of this procedure; namely, can different digit images from a different dataset still be directly useful? The surprising answer is yes. In the new version of the manuscripts we have tried to clarify this motivation. That said, you make a valid point in wanting to see less extreme versions with partial overlap in class labels.
> So we have added the second experiment, as described in “General Response to All Reviewers”.
>
> Comment: It is also particularly surprising to me that 25 MNIST samples were enough to train a LeNet to the point of 84% accuracy on the entire MNIST test set. (I'm referring to the target-only line) Is that really the case, or was a larger training set used for that particular line?
> Response: It is really the case that only 25 samples were used.  This 84% figure was taken directly from the paper Luo et al. 2017.  Having said that, we reproduced the experiment ourselves and got a very similar results - slightly below 84%.
>
> Comment: There is a claim that "SOSELETO has superior performance to all of the techniques which do not use unlabelled data", however I'm not sure whether these techniques were used as prescribed and if the comparison was fair.
> Response: All of the results (except for ours) are taken directly from the paper Luo et al. 2017.
>
> Comment: In related work, the "bilevel optimization" section could be a bit more descriptive, maybe some of the explanation given in Sect. 3 could be moved here?
> Response: Extra text has been added to Section 2 to explain bilevel optimization.
>
> Comment: The experiments were convincing, with the exception of the SVHN to MNIST section:
> Response: Please see “General Response to All Reviewers”.
>
> Comment: I believe a better synthetic experiment could be chosen to highlight the approach: how about a truly noisy dataset that is not as separable as the "noisy" dataset in Figure 1? Maybe you could have the same noisy dataset but with a small portion of random points having the wrong label. For the same experiment, it should be clearly stated that your task is binary classification and what was the classifier used.
> Response: Please see “General Response to All Reviewers”.
>
> Comment: For the CIFAR experiments, it is very good that it performs well, but it'd be informative to see if SOSELETO can perform even better with 10K samples.
> Response: If the paper is accepted, we will run this experiment for the camera ready copy.  For the moment, given that we were already better than the competition on CIFAR, we decided to focus our rebuttal efforts on re-doing the two other experiments, namely the SVHN to MNIST experiment with more digits and the revised, more illustrative synthetic experiment.
>
> Comment: It wasn't clear to me whether the a-values of only one batch (32 samples?) at a time were affected. If so, how does this scale to really large datasets like, say, Imagenet?
> Response: Yes, you are correct - only the alpha values of a given batch are affected.  To mitigate this, we use different sized batches for the source and target: the source batch (which determines the alpha values that are changed) can be considerably larger than the target batch; for example, in the case of CIFAR-10, we used a source batch of 256 vs. a target batch of 32.  Nevertheless, this is a limitation.  In practice, we were still able to get good performance in spite of this issue.
>
> Comment: In the CIFAR experiments, it is mentioned that a target batch-size is chosen to be larger to enable more a-values to be affected. This seems like a typo, but it was confusing. (I assume that the source batch-size is chosen to be larger)
> Response: You are correct, and this has been changed.
>
> Comment: It was not clear to me what "true transfer learning" means as opposed to domain adaptation.
> Response: Please see “General Response to All Reviewers”.
>
> Comment: This is a simple idea that seems to work well. As I wrote above, it would be great to know how it compares to other re-weighting techniques.
> Response: Please see “General Response to All Reviewers”.

---

### Author Response · Authors · 2018-11-23
**weights illustration - replaced the synthetic experiment and added an svhn to mnist with overlapping labels**

We thank all of the reviewers for their insightful comments.

*More Concrete Illustration of the Role of the Weights: Synthetic Experiment*
The reviewers clearly appreciated the crucial role of the source sample weights in SOSELETO.  However, they expressed a desire to see the role of these weights illustrated more clearly in the experiments.  We strongly agree with this comment, and have therefore inserted a new synthetic experiment in Section 4.1, including new figures and a revised explanation in the text. To summarize the results: the experiment has 500 points, of which 100 have incorrect labels; by thresholding the points according to their SOSELETO weights, we correctly classify 98.4% of the points in terms of whether they were correctly or incorrectly labelled.
Several new figures have been designed expressly to show the role of the weights:
- Given a particular weight threshold, we show which clean instances are incorrectly detected as noise, as well as which noisy instances are incorrectly detected as clean.
- We show the evolution of the range of weights on the noisy and clean instances as a function of training epochs.
- We show graphs of precision and recall vs. the weight threshold.
These new figures make clear the precise role of the weights in a synthetic setting.

*More Concrete Illustration of the Role of the Weights: SVHN to MNIST Experiment*
In order to further clarify the role of the weights in SOSELETO, we have added to the SVHN to MNIST experiment.  As SOSELETO is capable of automatically pruning useful instances at train time, the original experiment with zero-overlap in class labels was meant to demonstrate an extreme version of this procedure; namely, can different digit images from a different dataset still be directly useful? The surprising answer is yes. In the new version of the manuscripts we have tried to clarify this motivation. That said, the reviewers make a valid point in wanting to see less extreme versions with partial overlap in class labels.
Thus, we have added a second version of the experiment, in which the source dataset consists of all digits 0-9 of the SVHN dataset. The target dataset remains MNIST 5-9.
Our findings can be summarized as follows:
1. One immediate effect was an increase of the accuracy to above 90%.
2. Percentage of "good" instances (i.e. instances with weight above a certain threshold) didn't show very strong correlation with the labels.
3. That said, a more careful examination of low- and high-weighted instances, revealed that the usefulness of an instance, as determined by SOSELETO, is more tied to its appearance. Namely, whether the digit is centered, at a similar size as MNIST, the amount of blur, and rotation.
4. Especially interesting was that, among the "bad" (low-weighted) instances, we found about 3-5% wrongly labeled instances! The high weighted instances, on the other hand, had almost none.

We have included a summary of these results, along with illustrations in appendix C of the revised manuscript.

Note that we have still retained the original experiment (SVHN 0-4 to MNIST 5-9), so that we may continue to compare our results with existing techniques which report results for this specific experiment (see Table 2).

*Confusion re Terminology: Domain Adaptation vs. Transfer Learning*
At various points throughout the paper, the distinction between Domain Adaptation (DA) and Transfer Learning (TL) arises.  The main distinction we draw between these two regimes has to do with the overlap in the label sets of the source and target datasets.  In the case of DA, there is either full overlap between the label sets; or in some papers, partial but significant (50%).  We use the term TL to denote a situation in which there is either no overlap between label sets, or possibly very minimal overlap.  An example of the latter is the SVHN 0-4 to MNIST 5-9 experiment.  We realize that this distinction was perhaps not as clear as it might have been in the original submission.  We have therefore added a paragraph in Section 1 (“We pause here to note …”) to explain this more carefully.

*Clarification of Relationship to Existing Re-weighting Techniques*
In the original submission, one subsection of the Related Work section was dedicated to “Instance Re-Weighting”.  However, the reviewers have requested that the relationship to existing reweighting techniques be made clearer.  Crucially, the key distinction is that reweighting techniques are not applicable when source and target have non-overlapping label sets (the Transfer Learning regime, rather than the Domain Adaptation regime, as explained above).  Therefore, while related, existing instance re-weighting techniques solve a different problem than we do.
We have therefore added a paragraph in Section 1 (“We pause here to note …”) to make this distinction clearer.  We have also included some new references on the topic of reweighting in Section 2.

---

### Meta-Review · Area_Chair1 · 2018-12-16
**A simple approach for transfer learning but limited experimental evaluation**

**Confidence:** 5
**Recommendation:** Reject

**Metareview:**

The paper proposes an approach for transfer learning by assigning weights to source samples and learning these jointly with the network parameters. Reviewers had a few concerns about experiments, some of which have been addressed by the authors. The proposed approach is simple which is a positive but it is not evaluated on any of the regular transfer learning benchmarks (eg, the ones used in Kornblith et al., 2018 "Do Better ImageNet Models Transfer Better?"). The tasks used in the paper, such as CIFAR noisy -> CIFAR and SVHN0-4 -> MNIST5-9, are artificially constructed, and the paper falls short of demonstrating the effectiveness of the approach on real settings.

The paper is on the borderline with current scores and the lack of regular transfer learning benchmarks in the evaluations makes me lean towards not recommending acceptance.